# A Review of the Durability-Related Features of Waste Tyre Rubber as a Partial Substitute for Natural Aggregate in Concrete

**Yang Li [1],\*, Jiaqi Chai [1], Ruijun Wang [1], Yu Zhou [1] and Xiaogen Tong [2]**

1 State Key Laboratory of Eco-Hydraulics in Northwest Arid Region, Xi'an University of Technology, Xi'an 710048, China
2 China West Construction North Co., Ltd., Xi'an 710119, China
\* Correspondence: ly1990120311@163.com

**Abstract:** As the number of discarded tyres continues to increase, causing serious environmental problems, the need of recycling the waste tyre rubber become extremely urgent in worldwide. Today, there is an increasing focus on recyclable materials. The reuse of waste tyre rubber in concrete contributes to sustainable development. In the past 10 years, numerous experiments on the recovery of rubber from waste tyres to produce concrete products have been conducted. In this review, we conclude the major achievement of rubberized concrete (RC) durability, discuss and analyse the influence of rubber replacement rates, replacement patterns, particle size and treatment methods. Results show that an increase in rubber content can improve the chloride penetration resistance, acid and sulphate attack resistance, freeze–thaw resistance, and alkali–silica reaction damage resistance of concrete, and the content of 5–20% has a significant improvement effect. Rubber replacing fine aggregate is the best scheme for durability, followed by cement and coarse aggregate. In addition, the recommended rubber particle size is 0–3 mm. However, the rubber particle has adverse effects on abrasion resistance, impermeability, water absorption resistance and carbonation resistance. The pre-treatment of rubber or the addition of supplementary cementitious materials are effective and viable ways of improving the durability of RC. Further research is needed on the long-term durability of RC, as well as on ductility, energy absorption, and thermal and corrosion resistance.

**Keywords:** rubberized concrete; crumb rubber; durability property; rubber surface treatment; rubber size; rubber content

## 1. Introduction

Concrete is used extensively and widely in construction projects; however, it can be a constant drain on resources and energy. In the context of sustainable development, recycled concrete is receiving increasing attention [1]. Therefore, the researchers focus on producing concrete from sustainable materials. For example, the use of pozzolanic material to produce concrete has a low environmental impact while improving performance [2]. In addition, many types of waste can be recycled, including waste tyres [3], waste glass [4], waste lathes [5]. Fibres are often used to improve the performance of concrete [6]. Studies have shown that recycled steel fibres from waste tyres or lathes can be effective in enhancing the mechanical properties of concrete [7,8], and recycled rubber from waste tyres is also a promising sustainable building material.

Tyres have grown exponentially caused by the rapid population growth and economic development, and substantial waste tyres have been generated due to being beyond their service life [9,10]. Burning and landfilling are the two easiest and cheapest disposal methods. Burning rubber increases $CO_2$ emissions and releases toxic gases, such as styrene and butadiene, which seriously pollute the natural environment [11–13]. Rubber is difficult to decompose, and landfills seriously pollute the soil and make the land lose its vitality [14]. In recent years, recycling waste tyre to replace the cement or natural aggregates in concrete

is a promising research direction [15–17]. Studies have shown that recycled tyre rubber and steel fibres can improve the compressive, splitting tensile strength [18], dynamic mechanical properties [19], ductility, and energy absorption capacity of concrete [20]. The recycling of waste tyre in the above way prevents environmental pollution and reduces $CO_2$ emissions. Additionally, the use of natural aggregate in civil engineering can also be reduced, which is beneficial to protecting the ecological environment. This approach is resource-saving and environmentally friendly.

Rubber can replace some of the natural aggregates mixed into concrete, and the study found that rubber can significantly improve the dynamic mechanical properties of concrete [21]. However, the bonding ability between rubber aggregate (RA) and cement paste is weak, and microscopic analysis found micro cracks in the interface transition zone (ITZ) between RA and cement paste [22]. Therefore, rubber will reduce the compressive strength, tensile strength, and other mechanical properties of concrete [23]. Rubber can affect the workability of concrete due to its strong hydrophobicity [24]. Researchers have investigated the possibility of enhancing the adhesion of rubber particles by means of adhesives, activators, or other additives to effectively enhance the mechanical properties [25]. The durability of rubberized concrete (RC) can also be compromised due to the incompatibility of rubber with various concrete components. It has been found that pre-treatment of rubber can reduce the adverse effects on durability [26]. The rubber particle surface has been pretreated or precoated with NaOH [27], $KMnO_4$, and $NaHSO_3$ [28], cement mortar [29], silane coupling agent (SCA) [30], acrylic acid and polyethylene glycol [31], ethanol and acetone, carboxylated styrene–butadiene rubber (CSBR) latex [32,33] acids ($H_2SO_4$, HCl, $HNO_3$, $CH_3CHOOH$) [34–36], $CaCl_2$, and $H_2O_2$ [34], organic sulphur compounds [37], limestone powder (LP), silica fume (SF) [38,39], ethoxyline resin, and styrene-butadiene-type copolymer [40,41]. Table 1 summarizes some improvement methods for RC by modifying rubber particles.

Some reviews have been published on RC with the field of fresh and hardened properties [10,21,22,24], which are summarized in Table 2. However, in these reviews, there was a lack of detailed description of durability. Some important properties, including water absorption, chloride penetration resistance, carbonization resistance, alkali–silica reaction (ASR) damage resistance, acid resistance, and sulphate attack, were less analyzed. Furthermore, the RC reported in the review was limited to untreated crumb rubber (CR), and it did not cover reports on RC with physical and chemical treated CR. Additionally, the previous review did not analyze in detail the impact of rubber particle size, replacement rate and replacement pattern on durability. In addition, the existing reviews lack RC durability reports in recent years, and most of their coverage is research in 2018 and before. Therefore, it is necessary to carry out the latest comprehensive review of concrete containing waste rubber in order to better understand the durability of this important construction material.

This paper covers the study of various methods to enhance the durability of RC in the past 10 years and summarizes the rubber particles treatment methods to enhance the durability. The effects of rubber particle size, replacement rate, and surface treatment method on RC durability are compared. In accordance with the different durability, the survey results are summarized and compared with the control group. The review provides theoretical and experimental basis for dealing with waste tyre rubber in a green and environmental way.

Table 1. Comparison of various treatment methods of rubber waste on concrete.

| Treatment Method | Mechanism | Advantages | Disadvantages |
|---|---|---|---|
| Pre-treating with NaOH solution [27] | Remove zinc stearate from rubber particle surface, make rubber surface hydrophilic. | High efficiency, widely used, and synergy with other methods. | The compressive strength is not improved, even slightly reduced. |
| Pre-treating with KMnO$_4$ [28] | Oxidize rubber surface to make the surface hydrophilic. | High efficiency, cheap, and alternative chlorinated oxidizer. | Complicated operation and time consuming. |
| Pre-treating with NaHSO$_3$ [28] | Sulphonate rubber surface to make the surface hydrophilic. | High efficiency and cheap. | Complicated operation and time consuming. |
| Pre-coating with LP [38] | Make the rubber surface hydrophilic and rough. | Cheap and easy access to raw materials. | The void content and water absorption increase. |
| Pre-coating with ethoxyline resin [40] | Make the rubber surface hydrophilic and sticky. | High efficiency. | The freeze–thaw resistance decreases. |
| Pre-coating with emulsion [40] | Increase the rubber elastic modulus of and make the rubber hydrophilic. | High efficiency and simple operation. | The compressive strength and axial compressive strength decrease. |
| Pre-coating with styrene-butadiene-type copolymer [41] | Make the rubber hydrophilic and rough. | High efficiency. | The freeze–thaw resistance decreases slightly. |
| Pre-treating with SCA [30,33,39] | Facilitate the chemical bonding between rubber and cement. | High efficiency and easy to combine other methods. | The dynamic elastic modulus decreases slightly. |
| Pre-coating with cement [30,39] | Make the rubber hydrophilic and strengthen the elastic modulus. | High efficiency and easy access to raw materials. | Excessive cement coating is not conducive to the increase of density. |
| Pre-coating with Na$_2$SiO$_3$ [39] | Promoting the generation of calcium silicate hydrate gel in ITZ. | High efficiency and environment friendly. | The air content increases. |
| Pre-treating with acetone [32] | Facilitate the bonding between rubber and cement. | High efficiency and simple operation procedure. | The workability decreases. |

Table 2. Properties summarized in previous review studies.

| References | Workability | Density | Compressive Strength | Splitting Tensile Strength | Flexural Strength | Modulus of Elasticity |
|---|---|---|---|---|---|---|
| [10] | √ | √ | √ | √ | √ | √ |
| [21] | √ | √ | √ | √ | √ | √ |
| [22] | - | - | √ | √ | √ | √ |
| [24] | √ | √ | √ | √ | √ | √ |

| References | Abrasion Resistance | Water Absorption | Permeability | Freeze–Thaw Resistance | Acid Resistance | Sulphate Resistance |
|---|---|---|---|---|---|---|
| [10] | √ | - | - | - | - | - |
| [21] | √ | √ | √ | √ | - | - |
| [22] | - | - | √ | √ | √ | √ |
| [24] | √ | - | - | - | - | - |

| References | Chloride Penetration Resistance | Carbonation Resistance | Alkali–Silica Reaction Damage Resistance | Shrinkage | Microstructure | Long-Term Behavior |
|---|---|---|---|---|---|---|
| [10] | - | - | - | - | - | - |
| [21] | √ | √ | - | √ | - | - |
| [22] | - | - | - | √ | √ | - |
| [24] | - | - | - | - | √ | - |

## 2. Classification of Rubber Particles

Waste tyre rubber mainly comes from automobile and truck tyres. The rubber composition of the tyre is shown in Table 3. Although the composition of car and truck tyres is different in various countries, their natural and synthetic rubber content, which is approximately 40–45%, is similar [21]. Automobile tyres are produced more than truck tyres; hence, the rubber used in research mainly comes from automobile tyres to facilitate the waste tyre rubber recycling [42]. In previous research, six types of rubber have been used for recycling waste tyre rubber as building materials; the term name, particle size range, and type of replacement for natural aggregate are indicated in Table 4. Several commonly used tyre RAs with different sizes are shown in Figure 1. Rubber particles production is mainly achieved by using two methods, one is mechanical grinding, the other is cryogenic grinding [43], and then screening of rubber particles made from waste tyres to replace the fine or coarse natural aggregate [44,45]. Tyre rubber is ground into rubber powder or rubber ash to replace a certain percentage of cement, rubber fibre, and CR replace partial fine aggregate [46,47].

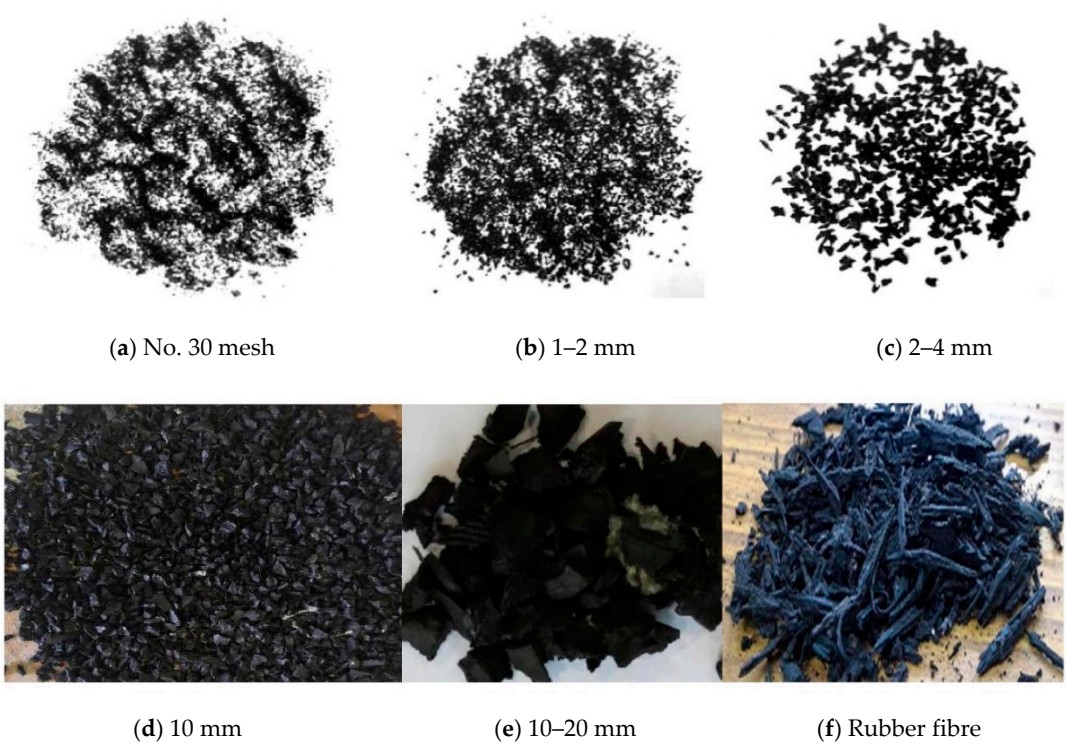

(**a**) No. 30 mesh          (**b**) 1–2 mm          (**c**) 2–4 mm

(**d**) 10 mm          (**e**) 10–20 mm          (**f**) Rubber fibre

**Figure 1.** Various sizes of tire particles [48].

**Table 3.** Composition of manufactured tires by weight [23,24,42].

| Materials | In USA | | In European Union | |
|---|---|---|---|---|
| | **Truck Tyre** | **Car Tyre** | **Truck Tyre** | **Car Tyre** |
| Natural rubber (%) | 27 | 14 | 30 | 22 |
| Synthetic rubber (%) | 14 | 27 | 15 | 23 |
| Carbon black (%) | 28 | 28 | 20 | 28 |
| Steel (%) | 14–15 | 14–15 | 25 | 13 |
| Others (textile, fillers, curatives, stabilizers, antioxidants, and antiozonants) (%) | 16–17 | 16–17 | 10 | 14 |

**Table 4.** Classification of the rubber particles [24,42].

| Term Name | Particle Size Range | Replacement Type |
|---|---|---|
| Shredded tyre | 100–230 mm in width, 300–460 mm in length | Coarse aggregate |
| Chipped tyre | 13–76 mm | Coarse aggregate |
| Fibre rubber | 2–5 mm in width, 10–22 mm in length | Fine aggregate |
| Granulated crumb rubber | 0.5–9.5 mm | Coarse aggregate or Fine aggregate |
| Crumb rubber | 0.425–4.75 mm | Fine aggregate or cement |
| Rubber powder/Rubber ash | ≤0.425 mm | Cement |

## 3. Abrasion Resistance

The abrasion resistance of concrete may be defined as its ability to resist being worn away by rubbing, which is important in the use of concrete structures [49,50]. There are two common methods for testing the abrasion resistance of concrete [51]. One is the "underwater method" specified by ASTM C1138 [52]. The other is the "ring method" named by the Chinese hydraulic concrete test code SL 352-2006 [53]. As shown in Table 5, previous experimental research has indicated that rubber that replaces natural aggregate in ordinary concrete reduces abrasion resistance. Bisht and Ramana [54] observed that as fresh RC vibrated, the rubber particles moved to the surface of the test piece and accumulated, causing the surface strength to decrease. Weak cementing ability between CR aggregate and cement paste made the RC prone to pulling and cracking, resulting in friction cracks. The increase of rubber content leads a decrease of RC abrasion resistance. Gupta et al. [49] observed that rubber ash mixed into the concrete resulting the compressive strength decrease, leading to poor abrasion resistance. An increase in cement matrix porosity and the weak ITZ between RA and cement paste take primary responsibility for decreased of RC abrasion resistance; the former is due to an increase in water–cement ratio (w/c) that ensures the workability of RC, whereas the latter is caused by the weak binding capacity of rubber particles [50]. Turki et al. [55] detected the inside of RC via scanning electron microscopy (SEM) and found that RA is easier to separate from cement matrix compared with natural aggregate. As shown in Figure 2, the siliceous aggregate is closely combined with the cement matrix; however, there are obvious cracks between RA and cement matrix. Some studies show that due to the weak bonding between RA and cement matrix, rubber particles are easy to rise to the surface during vibration, resulting in uneven distribution, which will reduce the abrasion resistance of concrete. In addition, poor ITZ will reduce the mechanical properties and affect the abrasion resistance [22]. Therefore, guaranteeing concrete strength, enhancing the binding capacity of RAs and reducing the RC porosity are necessary to ensure RC abrasion resistance. Adding supplementary cementitious materials (SCMs), for example ground-granulated blast-furnace slag (GGBFS), metakaolin (MK), SF, and fly ash (FA), into RC can effectively enhance the abrasion resistance; SCMs can enhance the concrete strength by filling the internal space. In addition, the additional hydration reaction strengthens the internal structure of the RC [51,56–59].

**Table 5.** Influence of rubber on the concrete abrasion resistance.

| Reference | Treatment Method | RA [a] Type and Size (mm) | RA Replacement Ratio (%) | Replacement Pattern | Concrete Type | Variation in Abrasion Resistance | D, ML, CL, ARS Compare to the Control Type (%) |
|---|---|---|---|---|---|---|---|
| Bisht and Ramana [54] | Untreated | CR: 0.6 | 4, 4.5, 5, 5.5 by weight | FAG | OC | ↓ [b] | 1.27% ↑, 7.59% ↑, 11.39% ↑, 17.72% ↑ (D) |
| Ridgley et al. [56] | Untreated | CR: 0–4.5 | 40 by volume | FAG | OC | ↓ | 55.71% ↑ (ML) |
| Thomas and Gupta [57] | Untreated | Rubber powder: 0.6 (40%) and CR: 0.8–2 (35%) + 2–4 (25%) | 2.5, 5, 7.5, 10, 12.5, 15, 17.5, 20 by weight | FAG | HSC | ↑ | 7.04% ↓, 19.72% ↓, 21.13% ↓, 19.72% ↓, 20.42% ↓, 27.46% ↓, 28.17% ↓, 32.39% ↓ (D) |
| Mohammed et al. [59] | Untreated | Rubber powder: 0.6 (40%) and CR: 1–3 (40%) + 3–5 (20%) | 10, 20, 30 by volume | FAG | RCC | ↓ | 10% ↓, 10% ↑, 12.5% ↑ (CL) |
| Shen et al. [60] | Untreated | CR: 1.18–4.75 | 18 by volume | CAG | Polymer modified porous concrete | ↑ | 13% ↓ (D) |
| Silva et al. [61] | Untreated | CR: 1.18–2.36 | 10, 20, 30, 40, 50 by weight | FAG | Paving block concrete | ↑ | 1.37% ↑, 8.22% ↓, 16.44% ↓, 12.33% ↓, 17.81% ↓ (ML) |
| Gesoğlu et al. [62] | Untreated | CR: 0.1–1 | 10, 20 by volume | CAG | Pervious RC | ↑ | 57.78% ↓, 80% ↓ (D) |
| Thomas et al. [63] | Untreated | Rubber powder 0.6 (40%) and CR 0.8–2 (35%) + 2–4 (25%) | 2.5, 5, 7.5, 10, 12.5, 15, 17.5, 20 by weight | FAG | OC | ↑ | 8.51% ↓, 12.77% ↓, 2.13% ↓, 12.06% ↓, 13.48% ↓, 15.60% ↓, 16.31% ↓, 15.60% ↓ (D) |
| Sukontasukkul and Chaikaew [64] | Untreated | CR: 1.2–5 | 10, 20 by weight | FAG and CAG | Pedestrian block concrete | ↓ | 303.3% ↑, 1376.7% ↑ (ML) |
| | Untreated | CR: 0.16–1.2 | 10, 20 by weight | FAG and CAG | Pedestrian block concrete | ↓ | 223.3% ↑, 973.3% ↑ (ML) |
| | Untreated | CR: 0.16–5 | 10, 20 by weight | FAG and CAG | Pedestrian block concrete | ↓ | 186.7% ↑, 756.7% ↑ (ML) |
| Gupta et al. [49] | Untreated | Rubber ash: 0.15–1.0 | 5, 10, 15, 20 by volume | FAG | OC | ↓ | 5.04% ↑, 8.4% ↑, 16.8% ↑, 19.3% ↑ (D) |
| He et al. [28] | Pre-treating with KMnO$_4$ and NaHSO$_3$ | Rubber powder: 0.425 | 2, 4, 6 by weight | FAG | OC | ↑ | 5.1% ↑, 17.9% ↑, 41.1% ↑ (ARS) |
| Onuaguluchi [38] | Pre-coating with LP | CR: 0.9–3 | 5, 10, 15 by volume | FAG | OC | ↑ | 8% ↓, 19.9% ↓, 11.7% ↓ (ML) |
| Segre [27] | Pre-treating with NaOH | Rubber powder: 0–0.5 | 10 by weight | FAG | OC | ↑ | 57.3% ↓ (ML) |

[a] Abbreviations: RA: Rubber aggregate, CAG: Coarse aggregate, FAG: Fine aggregate, D: Wear depth, ML: Mass loss, ARS: Abrasion resistance strength, CL: Cantabro loss, OC: ordinary concrete. [b] Evaluate standard change compared to the control type (%), corresponding to RA replacement ratio. Increase: ↑, Decrease: ↓.

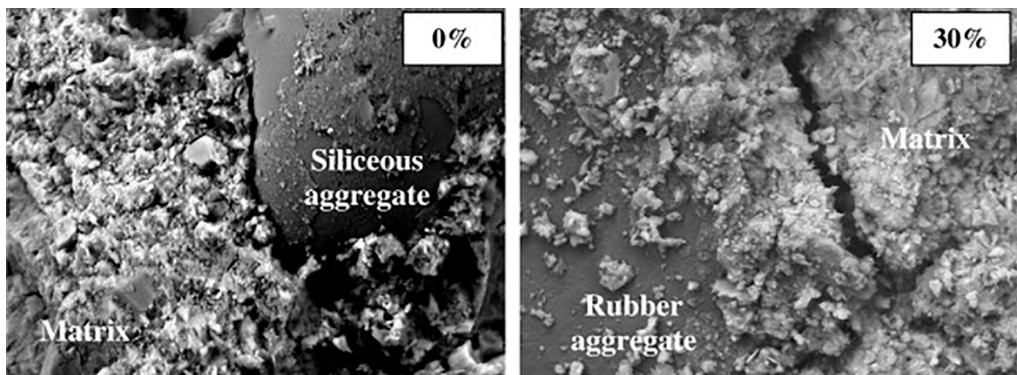

**Figure 2.** Adherence of ITZ of mortars with 0% RA and RA mortars with 30% RA [55].

Precoating or pretreating rubber particle surface before mixing into concrete can also improve the concrete abrasion resistance. Due to the hydrophobic property of RA, it can cause poor adhesion between cement paste and RA, which greatly reduces the mechanical properties, and limit the use of RC in a wide range. He et al. [28] reported that treatment of CR with $KMnO_4$ and $NaHSO_3$ solutions could make the surface of rubber particles highly hydrophilic, thus improving the binding capacity. Segre and Joekes [27] indicated that using NaOH to treat RA can enhance the rubber-matrix adhesion. Figure 3 show the state of contact surface with and without NaOH pretreatment. After treatment with NaOH, the crack width of ITZ is narrow, and the RA is difficult to detach [65]. Onuaguluchi [38] found that the two-stage approach of precoating rubber particles with LP and mixing SF into RC has also achieved a positive effect on the abrasion resistance. After LP precoating, the rubber surface becomes hydrophilic and acts as the reaction area of early hydration products. SF can generate a pozzolanic reaction on a rubber surface. The accumulation of hydration products on rubber surface changes the surface morphology and improves the binding capacity of RA [38]. Good binding capacity and ductility can make RA limit the formation and development of cracks caused by friction and reduce the stress concentration at the crack tip; hence, they can effectively improve abrasion resistance [66].

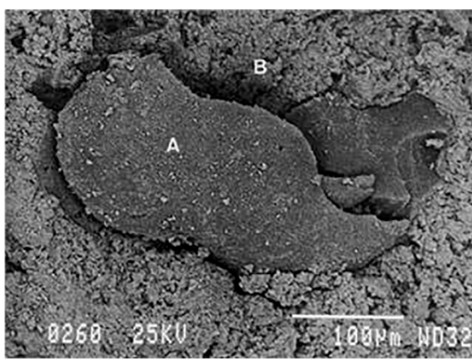

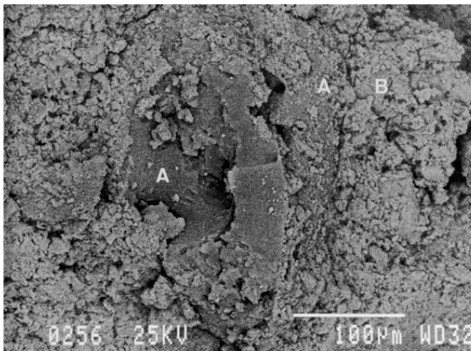

(**a**) With 10% by mass of as-received tire rubber

(**b**) With 10% by mass of NaOH-treated tire rubber

**Figure 3.** SEM images of fracture surface of cement test specimen. (**a**) Rubber particle; (**b**) Cement paste [27].

As shown in Table 5, the shape, size, and gradation of rubber particles also affect abrasion resistance. Fibre rubber particles have better ability to enhance abrasion resistance compared with granular rubber particles, which may be due to the fibre-holding effect preserving the integrity of cement paste [49,61]. The small particle size is better than that of larger one. Gesoğlu et al. [62] described that the combination of RA with different particle sizes can produce a good synergistic effect, Sukontasukkul and Chaikaew [64] also

observed the same phenomenon. Well-graded RA makes the inner part of RC considerably compact [63].

## 4. Water Absorption and Permeability

As shown in Table 6, as rubber content increases, water absorption also increases. This is mainly due to the increase of porosity after rubber is mixed into concrete [63]. The hydrophobic CR particle surface, during the mixing process, rubber particles trap air, resulting in increased internal pores between fine aggregate and CR, as shown in Figure 4 [67]. The light weight and low elastic modulus of RA cause the RC low compactness. It will be more difficult to fill pores and provide adequate compactness. Thus, the compaction of RC specimens becomes difficult and leads to an increase in internal porosity [61,66,68]. A high w/c results in an increase in porosity inside RC, because the RC needs a high mixing water content to ensure effective workability [69]. Measures are taken to reduce vibration time to prevent light weight rubber particles from floating and avoid separation during the vibration process. Consequently, porosity increases due to the lack of sufficient vibration compactness time. When cement paste hardens, many closed cavities are formed inside concrete, as shown in Figure 5. Concrete with 0% CR has a denser internal structure compared with concrete mixed with RA. In concrete with 4% and 5.5% CR, closed cavities are a good water storage bag, which provides the possibility of high water absorption. Figure 6 depicts that ITZ is a porous and weak structure due to the poor cementation ability of rubber particle surface; such structure easily causes RA separate from cement paste [69]. The compressive strength is significantly reduced with rubber addition, which easily produces micro cracks in hardened cement matrix. The cracks around the rubber and in the cement matrix connect the originally closed cavities to form a good water seepage and storage channel, which help to increased water absorption [70,71].

**Table 6.** Influence of rubber on the concrete water absorption.

| Reference | Treatment Method | RA Type and Size (mm) | RA Replacement Ratio (%) | Replacement Pattern | Concrete Type | Variation in Water Absorption | Water Absorption Compared to the Control Type |
|---|---|---|---|---|---|---|---|
| Youssf et al. [72] | Untreated | CR: 1.18 and 2.36 | 10, 20, 30, 40, 50 by volume | FAG | OC | ↑ [a] | 3.23% ↓, 12.90% ↑, 29.03% ↑, 32.26% ↑, 45.16% ↑ |
| Bisht and Ramana [54] | Untreated | CR: 0.6 | 4, 4.5, 5, 5.5 by weight | FAG | OC | ↑ | 12.57% ↑, 26.18% ↑, 41.88% ↑, 68.06% ↑ |
| Hunag et al. [73] | Untreated | CR: 0–4.75 | 10, 20, 30, 40 by volume | FAG | lightweight aggregate concrete | ↑ | 1.43% ↑, 14.29% ↑, 28.57% ↑, 35.71% ↑ |
| Thomas et al. [63] | Untreated | Rubber powder 0.6 (40%) and CR: 0.8–2 (35%) + 2–4 (25%) | 2.5, 5, 7.5, 10, 12.5, 15, 17.5, 20 by weight | FAG | OC | ↑ | 16.67% ↑, 16.67% ↑, 33.33% ↑, 66.67% ↑, 66.67% ↑, 83.33% ↑, 100.00% ↑, 133.33% ↑ |
| Onuaguluchi and Panesar [68] | Untreated | CR: 0–2.3 | 5, 10, 15 by volume | FAG | OC | ↑ | 12.73% ↑, 14.55% ↑, 27.27% ↑ |
| Benazzouk et al. [74] | Untreated | CR: 0–1.0 | 10, 20, 30, 40, 50 by volume | Cement | OC | ↓ | 52.33% ↓, 68.39% ↓, 74.61% ↓, 77.20% ↓, 80.83% ↓ |
| Mohammed et al. [67] | Untreated | CR: 0.1–0.5 | 10, 25, 50 by volume | FAG | OC | ↑ | 5% ↑, 20% ↑, 45% ↑ |
| Gesoğlu and Güneyisi [75] | Untreated | CR: 0.15–2.0 | 5, 15, 25 by volume | FAG | OC | ↓ | 2.4% ↓, 6.2% ↓, 1.8% ↓ |
| Mohammed and Adamu [59] | Untreated | CR: 0.6 (40%), 1–3 (40%), 3–5 (20%) | 10, 20, 30 by volume | FAG | OC | ↓ | 22.5% ↓, 13.4% ↓, 5.8% ↓ |

**Table 6.** *Cont.*

| Reference | Treatment Method | RA Type and Size (mm) | RA Replacement Ratio (%) | Replacement Pattern | Concrete Type | Variation in Water Absorption | Water Absorption Compared to the Control Type |
|---|---|---|---|---|---|---|---|
| Girskas and Nagrockienė [71] | Particle size effect | CR: 2–4 | 5, 10, 20 by weight | FAG | OC | ↑ | 14.04% ↑, 30.66% ↑, 41.83% ↑ |
| Thomas and Gupta [57] | Particle size effect | Rubber powder: 0.6 (40%) and CR: 0.8–2 (35%) + 2–4 (25%) | 2.5, 5, 7.5, 10, 12.5, 15, 17.5, 20 by weight | FAG | OC | ↑ | 1.52% ↓, 4.55% ↓, 4.55% ↓, 3.03% ↓, 0%, 3.03% ↑, 7.58% ↑, 12.12% ↑ |
| Sukontasukkul and Tiamlom [76] | Particle size effect | CR: 0.5–3.35 | 10, 20, 30 by volume | FAG | OC | ↑ | 15.38% ↑, 19.23% ↑, 42.31% ↑ |
| Ganjian et al. [42] | Particle size effect | Chipped rubber: 2.5–11.0 | 5, 7.5, 10 by weight | CAG | OC | ↑ | 2.38% ↑, 45.24% ↑, 64.29% ↑ |
| Benazzouk et al. [74] | Particle size effect | Rubber powder: 0–1.0 | 10, 20, 30, 40, 50 by volume | Cement | OC | ↓ | 52.33% ↓, 68.39% ↓, 74.61% ↓, 77.20% ↓, 80.83% ↓ |
| Li et al. [77] | Particle size effect | Rubber powder: 0–0.3, 1–2 | 30 by volume | FAG | SCC | ↓ | 10.9% ↓, 24.4% ↓ |
| Kashani et al. [78] | Pre-treating with NaOH, | CR: 0.9–2.5 | 10, 20, 30 by weight | Total solid mass | Light weight cellular concrete | ↓ | 14.29% ↓, 60.71% ↓, 66.07% ↓ |
| Meddah et al. [79] | Pre-treating with NaOH, followed pre-coating with resin | Granulated CR: 2.5–5 | 5, 10, 15, 20, 25, 30 by volume | CAG | RCC | ↓ | 16.46% ↓, 41.77% ↓, 49.37% ↓, 60.76% ↓, 63.29% ↓, 70.89% ↓ |
| Onuaguluchi [38] | Pre-coating with LP | CR: 0.9–3 | 5, 10, 15 by volume | FAG | OC | ↓ | 23.2% ↓, 43.4% ↓, 49.9% ↓ |

[a] Evaluate standard change compared to the control type (%), corresponding to RA replacement ratio. Increase: ↑, Decrease: ↓.

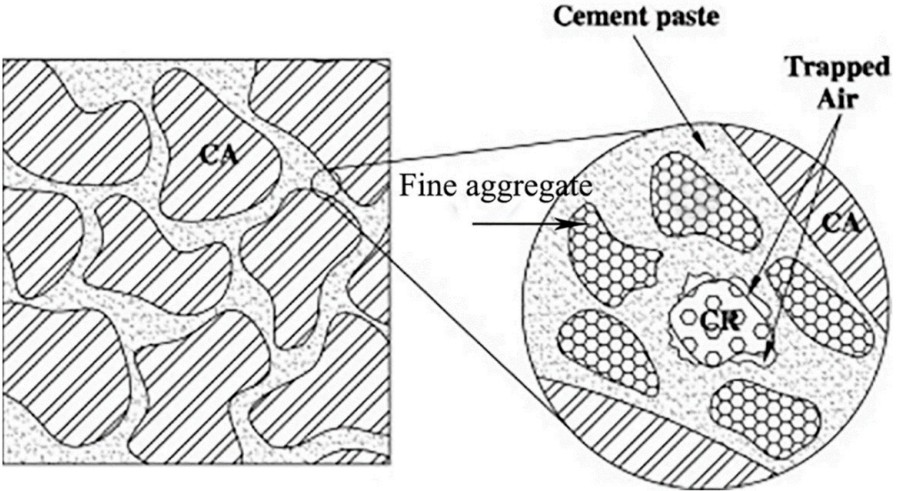

**Figure 4.** Microstructure of CR concrete [67].

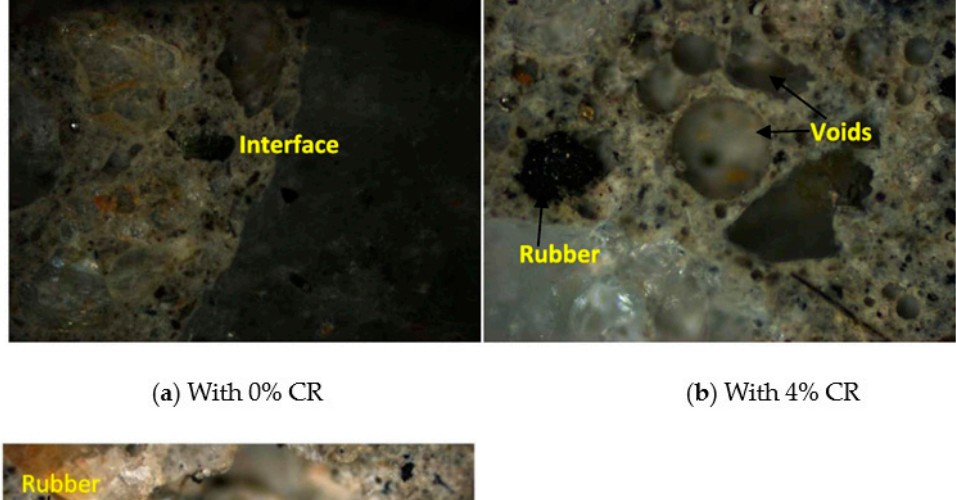

(a) With 0% CR　　　　　　　　　　　　　　　　　　　(b) With 4% CR

(c) With 5.5% CR

**Figure 5.** Optical microscope of CR concrete [54].

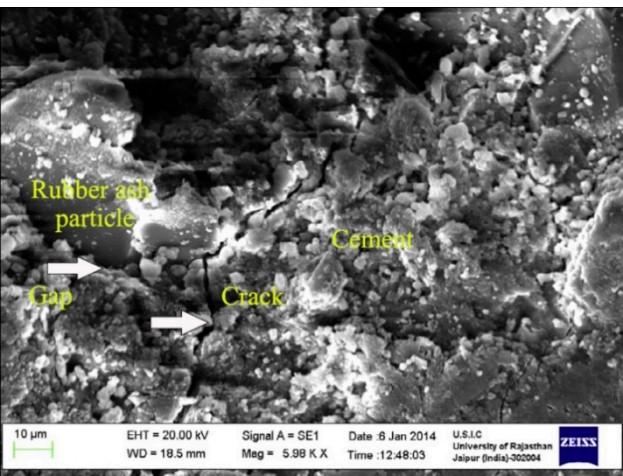

**Figure 6.** Microstructure of RC containing 20% rubber ash [49].

The influences of the rubber replacement pattern and ratio on the water absorption of RC are shown in Table 6. Youssf et al. [72] described that CR (1.18–2.36 mm) partially as the replacement of fine aggregate. When the CR content is less than 10%, the water absorption of RC decreases with the rubber content increases. However, when the CR content is more than 10%, the opposite results occur. A total of 0–10% of fine rubber particles can effectively fill the internal pores and reduce the internal porosity of concrete owing to the rubber particle filling effect. The external water can be prevented from infiltrating into the internal pores of concrete due to the hydrophobicity of rubber particles. However, when excessive rubber (10–20%) is added, the bonding capacity between the cement paste

and RA is poor and the internal cracks of RC increase, leading to an increase in water absorption [57,80]. Foamed concrete with 2% and 4% rubber contents is shown in the SEM diagram in Figure 7. Figure 7a illustrates that rubber particles fill in permeability channels, blocking the entry passage of water and improving the waterproof performance of foam concrete. As shown in Figure 7b, excessive rubber content leads to a dispersed cement matrix in foamed concrete; the poor cementation increases permeability channels and water absorption [81]. Thomas and Chandra Gupta [57] found that using three different particle size mixtures of CR (rubber powder 0.6 mm (40%) + CR 0.8–2 mm (35%) + CR 2–4 mm (25%) had a better filling effect compared with using a single-sized rubber alone.

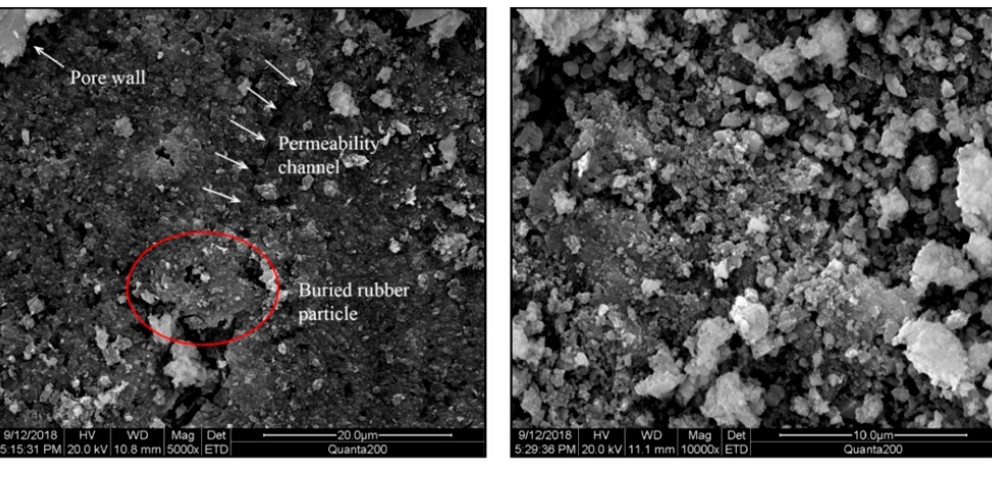

(**a**) Specimen with 2% CR content          (**b**) Specimen with 4% CR content

**Figure 7.** SEM images of foamed concrete incorporated with CR [81].

Rubber particles size also has great influence on the water absorption of concrete. Yilmaz and Degirmenci [82] incorporated 20% of three types of rubber particles with different particle sizes (0–0.25, 0.25–0.5, and 0.5–1.0 mm) into concrete, and the water absorption was 30%, 26%, and 24%. As the rubber particle size increased, the water absorption decreased slightly. Li et al. [77] found the same phenomenon. RC with fine RA needs substantial water to achieve the same workability as the control group; rubber particles with large particle size are easy to mix because the surface is smooth and spherical [71]. However, an excessively large particle size has a negative effect. Sukontasukkul and Tiamlom [76] reported that when CR No.6 (0.5–3.35 mm) was incorporated into concrete; as the blending amount increased, the water absorption increased. By contrast, CR No.26 (0–0.5 mm) incorporated into concrete reduced water absorption. The rubber surface formed bubbles due to the non-polar rubber particles surface, and the internal porosity of RC increased. The volume of the captured bubble was larger, and the RA size increased, as shown in Figure 8. Although the surface of CR No.26 also formed bubbles, the volume of the bubbles was small. These small bubbles formed a separate closed space and did not act as a water seepage channel. Moreover, CR No.26 had a good filling effect, and can effectively fill the pores and reduce the porosity in RC. Similar results were obtained with rubber powder (0.1–1 mm) as a replacement for partial cement in concrete [74].

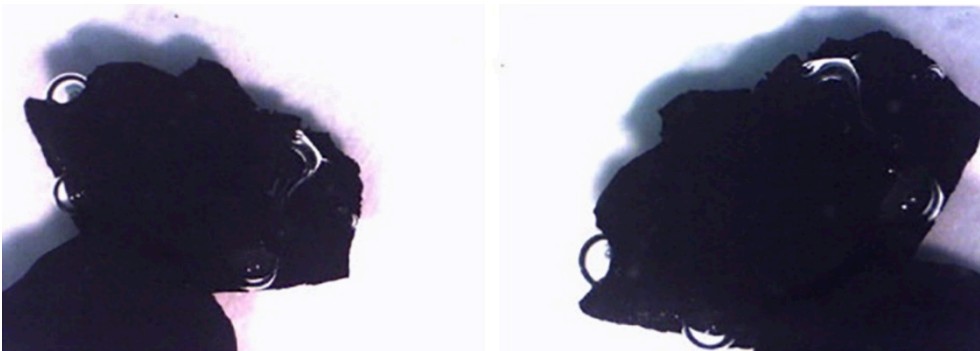

**Figure 8.** No. 6 CR trapped air bubbles in water [76].

The major approaches to reduce the water absorption are to strengthen the cementation ability between cement paste and RA, reduce the internal porosity, and reinforce the internal compactness of RC [83,84]. The rubber particles surface pretreated with NaOH can effectively strengthen the cementation ability and reduce the porosity due to the weak ITZ [79]. Adding SF to RC can effectively reduce water absorption. SF promotes hydration reaction, thereby increasing hydration production. Hydration products are filled in the pores amongst cement matrix, rubber, and aggregate, effectively reducing the internal porosity of concrete [38]. The addition of SF can react with the NaOH in the pores of ITZ, make the internal structure compact and strengthen the cementation ability [67]. Mixing FA, MK, or nanosilica into RC also produces the same effect as SF incorporation [59,75,80,85,86].

Permeability is an important factor affecting the durability of concrete, and it directly influences freeze–thaw resistance, chloride penetration, and acid and sulphate resistance. Permeability reflects the ability of fluid to pass through concrete, and the depth of water penetration is adopted as the standard to evaluate permeability. Table 7 indicates that as the rubber content increases, the permeability increases. The factors affecting the permeability of RC are similar to those of water absorption, which can be owed to the porosity increase of the internal structure and the increase of internal macro- and micro cracks [63]. Rubber particles mixed in concrete easily float in the cement matrix owing to the light weight and elastic properties of RA; this condition causes poor compaction of RC, resulting in increased porosity and water permeability in RC [73,87]. The weak cementation ability between cement matrix and RA leads to cracks around rubber particles [42]. Cracks around rubber particles and in cement make it easy for external water to penetrate concrete and transfer in various parts [54]. Su et al. [87] reported that rubber particles with small particle size (0–0.5 mm) can reduce permeability better than rubber particles with large particle size (0.5–3 mm) as are placement for fine aggregate. The better filling effect of small rubber particles in comparison with that of large particles causes RC to have a dense internal structure [88]. RA with good gradation and composed of different sizes of rubber particles is compact because fine rubber particles can effectively fill the gap formed by large rubber particles [87]. Si et al. [89] observed that rubber particles pre-treated with NaOH as a replacement for fine aggregate in SCC can effectively reduce permeability. RC needs a high w/c to achieve the required workability; a high w/c ratio causes a bad influence on permeability resistance [63]. Gupta et al. [90] observed that when the temperature exceeded 150 °C, a pore network would be formed in RC due to rubber fibre decomposition. The depth of water penetration reached 49 mm, which resulted in the high water penetration of RC. When RC has impermeability requirements, it is unsuitable for working in a high-temperature environment.

**Table 7.** Influence of rubber on the concrete water permeability.

| Reference | Treatment Method | RA Type and Size (mm) | RA Replacement Ratio (%) | Replacement Pattern | Concrete Type | Variation in Water Permeability | Permeability Compared to the Control Type |
|---|---|---|---|---|---|---|---|
| Thomas et al. [63] | Untreated | Rubber powder: 0.6 (40%) and CR: 0.8–2 (35%) + 2–4 (25%) | 2.5, 5, 7.5, 10, 12.5, 15, 17.5, 20 by weight | FAG | OC | ↑ [a] | 18.42% ↑, 13.16% ↑, 60.53% ↑, 105.26% ↑, 115.79% ↑, 115.79% ↑, 163.16% ↑, 163.16% ↑ |
| Hunag et al. [73] | Untreated | CR: 0–4.75 | 10, 20, 30 by volume | FAG | OC | ↑ | 16.67% ↑, 50.00% ↑, 66.67% ↑ |
| Bisht and Ramana [54] | Untreated | CR: 0.6 | 4, 4.5, 5, 5.5 by weight | FAG | OC | ↑ | 2.56% ↑, 7.69% ↑, 8.97% ↑, 19.23% ↑ |
| Thomas and Gupta [57] | Untreated | Rubber powder: 0.6 (40%) and CR: 0.8–2 (35%) + 2–4 (25%) | 2.5, 5, 7.5, 10, 12.5, 15, 17.5, 20 by weight | FAG | OC | ↑ | 0, 25% ↑, 75% ↑, 75% ↑, 100% ↑, 150% ↑, 150% ↑, 225% ↑ |
| Wang et al. [89] | Pre-treating with NaOH | CR: 1.44–2.83 | 15, 25 by volume | FAG | SCC | ↓ | 52.04% ↓, 54.08% ↓ |

[a] Evaluate standard change compared to the control type (%), corresponding to RA replacement ratio. Increase: ↑, Decrease: ↓.

## 5. Freeze–Thaw Resistance

As shown in Table 8, RAs replace natural aggregates in concrete can enhance the freeze–thaw resistance. As the rubber content increases, the freeze–thaw resistance becomes better. Turgut and Yesilata [91] used CR (particle size range: 0.075–4.75 mm) to replace fine aggregate in approximately 10% volume; after a freeze–thaw cycle test, it was 66.48% lower in the mass loss compared to the control group, the freeze–thaw resistance reached best with 30% CR content. Zhang et al. [89] pretreated CR with NaOH and $Na_2SiO_3$ mixed solution, it was found that using the treated CR instead of fine aggregate significantly improved the freeze–thaw resistance. When the replacement ratio was 5%, 10%, 15%, and 20%, the mass loss was reduced by 43.48%, 84.78%, 54.35%, and 71.74%, respectively. Rubber can be regarded as an air-entraining agent, making concrete achieve the same freeze–thaw resistance as air-entrained concrete [92]. Water enters into concrete under the action of hydraulic pressure and freezes at low temperature, expanding in volume, which is the major reason for the freeze–thaw damage [93,94]. The hydrophobic nature and small jagged shapes of rubber particle surface, as shown in Figure 9, tend to cause it to entrap air during the mixing process of RC, and an increase in ineffective pores improves the freeze–thaw resistance. When water is frozen and expanded in cement, effective pores can play the role of stress absorption mechanism, providing buffer space for water volume expansion [71]. The low rubber particles elastic modulus can absorb the volume expansion stress of water [95–97].

**Table 8.** Influence of rubber on the concrete freeze–thaw resistance.

| Reference | Treatment Method | RA Type and Size (mm) | RA Replacement Ratio (%) | Replacement Pattern | Concrete Type | The Ways of Freezing and Thawing | Freeze–Thaw Resistance | ML [a], RDME, SL, DF, FTRG Compared to the Control Type |
|---|---|---|---|---|---|---|---|---|
| Turgut and Yesilata [91] | Untreated | CR: 0.075–4.75 | 10, 20, 30, 40, 50, 60, 70 by volume | FAG | OC | Freezing (−9 °C) and thawing (25 °C) in water | ↑ [b] | 66.48% ↓, 66.11% ↓, 71.30% ↓, 68.52% ↓, 59.63% ↓, 69.26% ↓, 70.37% ↓ (ML) |
| Gonen [98] | Untreated | CR: 0.125–1 | 0.5, 1, 2, 4 by volume | FAG | OC | Freezing and thawing in 3% NaCl solution freezing | ↑ | 34.55% ↓, 40.91% ↓, 73.64% ↓, 87.27% ↓ (ML) |
|  | Untreated | CR: 0.25–2 | 0.5, 1, 2, 4 by volume | FAG | OC | Freezing and thawing in 3% NaCl solution freezing | ↑ | 38.18% ↓, 61.82% ↓, 70.91% ↓, 81.82% ↓ (ML) |
| Al-Akhras and Smadi [99] | Untreated | Rubber ash: 0–0.15 | 5, 10 by weight | FAG | OC | Freezing in air and thawing in water | ↑ | 211.11% ↑, 400.00% ↑ (DF) |
| Topçu and Bilir [100] | Untreated | CR: 0–4 | 3.7, 7.3, 10.98 by weight | FAG | OC | Freezing and thawing in water | ↓ | 37.36% ↑, 114.82% ↑, 170.27% ↑ (SL) |
| Zhu et al. [93] | Untreated | CR: 0.25 | 0.26, 0.5, 1.6 by weight | CAG and FAG | OC | Freezing (−15 °C) and thawing (6 °C) in water | ↑ | 75.00% ↑, 100.00% ↑, 87.50% ↑ (FTRG) |
| Paine [92] | Untreated | CR: 0.5–1.5 | 2, 4, 6 by volume | FAG | OC | Freezing and thawing in water | ↑ | 86.36% ↑, 75.00% ↑, 68.18% ↑ (RDME) |
| Liu et al. [40] | Pre-coating with synthetic resin | CR: 2.0–4.0 | 5 by volume | FAG | OC | Freezing (−16 °C) and thawing (3 °C) in water | ↑ | 1.2% ↓ (SL) |
| Si et al. [101] | Pre-treating with NaOH | CR: 1.44–2.83 | 15, 25, 35, 50 by volume | FAG | OC | Freezing (−18 °C) in air and thawing (4 °C) in water | ↑ | 4.07% ↑, 0.19% ↑, 1.74% ↓, 3.10% ↓ (RDME) |
| Pham et al. [41] | Pre-coating with styrene-butadiene-type copolymer | CR: 0.65–3 | 30 by volume | FAG | mortar | Freezing (−18 °C) and thawing (4 °C) in water | ↑ | 58.33% ↑ (RDME) |
| Wang et al. [102] | Pre-treating with NaOH | CR: 0.6–2.8 | 10, 15 by volume | FAG | OC | Freezing and thawing in water | ↑ | 8.02% ↑, 0.10% ↑ (DF) |
| Zhang et al. [103] | Pre-treating with NaOH and $Na_2SiO_3$ | CR: 0.15–4.75 | 5, 10, 15, 20 by volume | FAG | OC | Freezing (−20 °C) and thawing (5 °C) in water | ↑ | 43.48% ↓, 84.78% ↓, 54.35% ↓, 71.74% ↓ (ML) |

[a] Abbreviations: ML: Mass loss, RDME: Relative dynamic modulus of elasticity, SL: strength loss, DF: Durability factor, FTRG: Freezing–thawing resistance grade. [b] Evaluate standard change compared to the control type (%), corresponding to RA replacement ratio. Increase: ↑, Decrease: ↓.

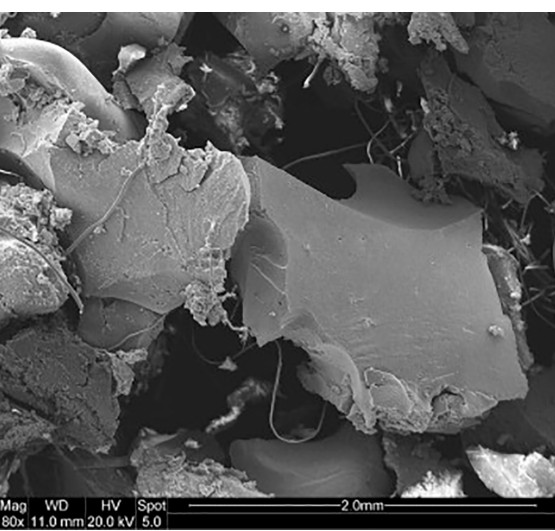

**Figure 9.** CR particle at 80× magnification [104].

According to the Table 8, it can conduct that the concrete with smaller particle size CR have higher freeze–thaw resistance. Gesoğlu et al. [62] used rubber with different particle sizes (<0.5, 1–1.5 and 1.5–2 mm) to replace fine aggregate and found that the best frost resistance occurs when the rubber particle size <0.5 mm. Zhu et al. [93] reported that a 60-mesh (0.25 mm) rubber particle achieved the best freeze–thaw resistance. In case of maintaining rubber content, the larger the rubber particle size, the farther the distance amongst the rubber particles, such relationship is unbeneficial to the freeze–thaw resistance. Small-sized rubber particles has a high specific surface area, which is conducive to air entrance. These micro-scale pools can serve as a buffer space for water freezing in cement [62]. Large-sized rubber particles are unbeneficial to the combination of rubber and cement paste, resulting in substantial cracks inside concrete, which make it easy for water to penetrate concrete [103]. Additives are often used to enhance the freeze–thaw resistance of concrete [105]. In addition, pretreatment of rubber can also be effective in improving freeze–thaw resistance. Using NaOH to treat the RA and then incorporating it into concrete had better freeze–thaw resistance in comparison with RC than RA without NaOH treatment [101]. Liu et al. [40] used six modifiers to precoat the rubber particles' surface and found that the precoating of synthetic resin could minimize the degree of mechanical strength reduction and achieve good freeze–thaw resistance. Precoating rubber particle with styrene-butadiene-type copolymer can better the bonding ability and effectively enhance the RC freeze–thaw resistance [41]. The negative effect of RA on concrete strength makes RC suitable for areas which do not have high strength requirements but need high freeze resistance, such as roads that need to undergo repeated freeze–thaw cycles in alpine regions [106]. However, the rubber mixed into SCC weakens the freeze–thaw resistance [100].

## 6. Acid and Sulphate Resistance

Acid solutions, such as $H_2SO_4$ or hydrochloric acid (HCl), penetrate concrete and react with $Ca(OH)_2$ and calcium silicate hydrate (CSH) gel, causing physical and chemical damages [107]. As shown in Table 9, the rubber incorporated into concrete can enhance the resistance to $H_2SO_4$ attack. The resistance to $H_2SO_4$ corrosion of RC effected by different rubber contents are shown in Figure 10 and evaluated by changes in compressive strength after soaking in 3% $H_2SO_4$ solution. In the early 7 days of immersion, the initial concrete internal hydration reaction, causing the formation of ettringite and CSH, which can increase the compressive strength. After 180 days of immersion, a minimum compressive strength loss occurred at 5.5% CR, which indicated that the RA relieved the structural damage of concrete caused by $H_2SO_4$ corrosion [108]. Gupta et al. [107] found that rubber reacted

with acid can enhance the rubber particles surface adhesion ability, which could exert the rubber bridge effect to suppress the generation of internal cracks and the separation of materials. Moreover, the rubber could alleviate the compressive stress generated by the expansion of the ettringite existing inside the cracks around the rubber particles [109]. HCl is less aggressive than $H_2SO_4$, as shown in Table 9. Gupta et al. [107] demonstrated that rubber incorporated into concrete enhanced resistance to HCl attack, and the combination of rubber fibre and rubber powder had improved results. Therefore, RC could be applied to an environment where acid corrosion occurs. Although high rubber content produces high resistance to $H_2SO_4$ attack, it can result in a reduction in the compressive strength of the concrete. Thomas et al. [110] observed that 4% CR satisfied serviceable strength and had good resistance to $H_2SO_4$ attack.

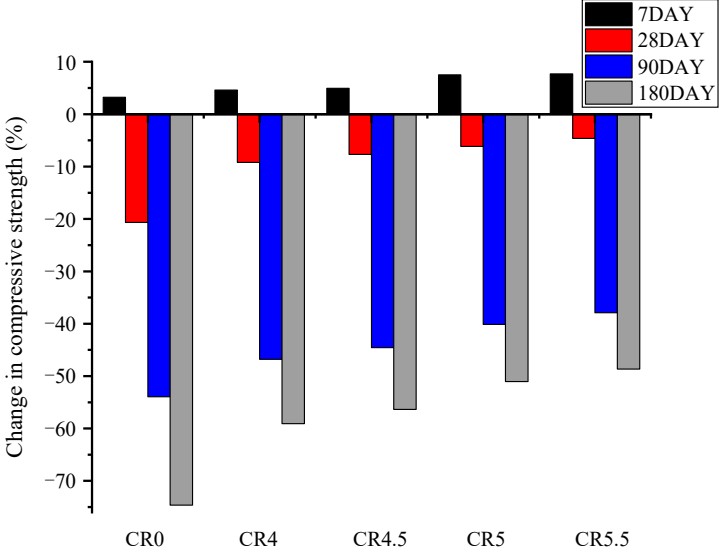

**Figure 10.** Change in compressive strength after soaking in 3% $H_2SO_4$ solution [108].

However, Rahimi et al. [111] reported that rubber incorporated into concrete reduced the resistance to $H_2SO_4$ attack, which is contrary to conclusions of previous research. Azevedo et al. [86] found that adding rubber to HPC greatly reduced the $H_2SO_4$ resistance. This phenomenon may be due to the increase in concrete permeability with the addition of rubber particles [112]. Using rubber fibre or rubber particles mixed with polyethylene terephthalate as fine aggregate could substantially increase the resistance to $H_2SO_4$ attack [111].

Sulphate reacts with hydration products to produce ettringite, which expands in volume to generate internal compressive stress that forms cracks inside concrete and causes the internal structure of cement to peel off and cause structural damage [108]. Table 9 shows that using RA to replace part of natural aggregate can enhance the resistance to sulphate attack of concrete. In accordance with previous experience, the content and size of rubber particles affect sulphate resistance. Using CR (0–4.7 mm) to replaced fine aggregate partially with different substitution levels (10%, 20%, 30%, and 40% by volume); after sulphate attack, mass loss was reduced by 7.02%, 3.51%, 1.75%, and 22.81%, respectively, compared with the control group [73]. With the rubber content increases, the ability of concrete to resist sulfate attack increases. Figure 11 depicts the resistance to sulphate attack in self-consolidating rubberized concrete (SCRC) effected by particle sizes (#30: 0.6 mm, #50: 0.3 mm) and different rubber contents. The weight loss of RC is less than that of the control group in the rubber content of 5–10%. When the content exceeds 10%, the opposite situation occurs. The best resistance to sulphate attack is achieved when 5% rubber powder (passes through a #30 sieve) is incorporated [117].

**Table 9.** Influence of rubber on the concrete acid and sulphate resistance.

| Reference | Treatment Method | RA Type and Size (mm) | RA Replacement Ratio (%) | Replacement Pattern | Concrete Type | HA [a], SA, and S Resistance | ML, EX, and ACC Compared to the Control Type |
|---|---|---|---|---|---|---|---|
| Thomas et al. [110] | Untreated | CR: 2–4 (25%) + 0.8–2 (35%) and rubber powder: 0.6 (40%) | 5, 10, 15, 20 by volume | FAG | OC | ↑ [b](SA) | 2% ↓, 10.47% ↓, 11.41% ↓, 14.82% ↓ (ML) |
| Thomas et al. [113] | Untreated | CR: 2–4 (25%) + 0.8–2 (35%) and rubber powder: 0.6 (40%) | 5, 10, 15, 20 by volume | FAG | OC | ↑(SA) | 0.49% ↑, 0.61% ↓, 21.17% ↓, 22.26% ↓ (ML) |
| Azevedo et al. [86] | Untreated | CR: 1–2.4 | 5, 10, 15 by weight | FAG | HPC | ↓(SA) | 7.41% ↑, 33.33% ↑, 50% ↑ (ML) |
| Gupta et al. [107] | Untreated | Rubber powder: 0.15–1.9 | 5, 10, 15, 20 by volume | FAG | OC | ↑(HA) | 1.11% ↓, 3.33% ↓, 4.67% ↓, 5.00% ↓, (ML) |
| | Particle size effect | Rubber fibres: width of 2–5, length up to 20, and rubber powder (10%) | 5, 10, 15, 20, 25 by volume | FAG | OC | ↑(HA) | 2.25% ↓, 4.49% ↓, 5.62% ↓, 4.50% ↓, 3.37% ↓ (ML) |
| Hunag et al. [73] | Untreated | CR: 0–4.7 | 10, 20, 30, 40 by weight | FAG | Low-strength lightweight aggregate concrete | ↑(S) | 7.02% ↓, 3.51% ↓, 1.75% ↓, 22.81% ↓ (ML) |
| Onuaguluchi and Banthia [114] | Untreated | CR: 0.2–2 | 10, 15 by volume | FAG | OC | ↑(S) | 56.84% ↓, 52.63% ↓ (EX) |
| Thomas et al. [115] | Untreated | CR: 2–4 (25%) + 0.8–2 (35%) and rubber powder: 0.6 (40%) | 5, 10, 15, 20 by volume | FAG | HSC | ↓(S) | 18.78% ↑, 74.11% ↑, 113.71% ↑, 154.31% ↑ (ML) |
| Liu et al. [40] | Untreated | CR: 0.2–4 | 5, 10, 15, 20 by volume | FAG | OC | ↑(S) | 0.62% ↑, 1.35% ↑, 1.66% ↑, 2.39% ↑ (ACC) |
| | Pre-coating with synthetic resin | CR: 2–4 | 5 by volume | FAG | OC | ↑(S) | 1.84% ↑ (ACC) |
| Li et al. [116] | Pre-treating with NaOH | CR: 0.85–2 | 10 by volume | FAG | OC | ↑(S) | 31.01% ↓ (ML) |

[a] Abbreviations: ML: Mass loss, EX: Expansion, ACC: Anti-corrosion coefficient, HA: Hydrochloric acid, SA: Sulphuric acid, S: Sulphate. [b] Evaluate standard change compared to the control type (%), corresponding to RA replacement ratio. Increase: ↑, Decrease: ↓.

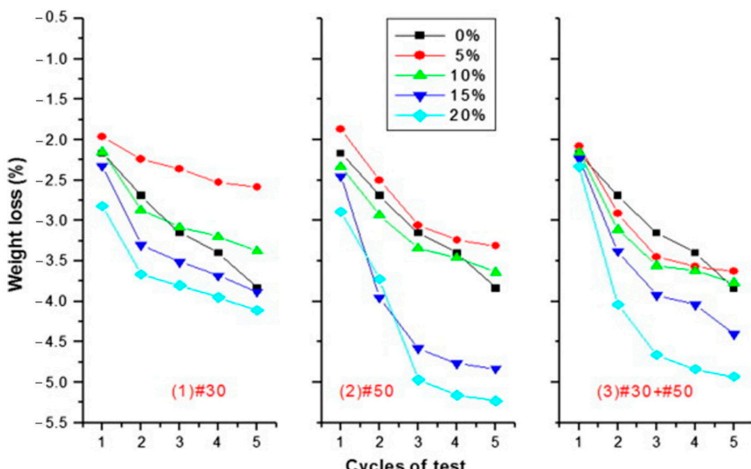

**Figure 11.** Weight loss after erosion with sodium sulphate solution for SCRC [117].

The incorporation of rubber into concrete can improve the resistance to sulphate attack, it can be attributed to the fact that compared with concrete without RA, the RC has better deformation ability, which can relieve the internal expansion stress caused by sulphate attack. Moreover, rubber particles can absorb expansion stress, prevent cracks from developing, and reduce the width of the cracks, thereby reducing the intrusion of sulphate ions into the interior of concrete [116]. The incorporation of fine rubber particles increases the micro-sized void content, which can act as the buffer space for the volume expansion of gypsum and ettringite [114]. Onuaguluchi and Banthia [114] reported that the RC incorporated with 10% SF gains great resistance to sulphate attack. This condition can be attributed to the reduced aluminate/$Ca(OH)_2$ in the matrix and the compact interior of the RC; the intrusion of sulphate ions becomes difficult. Li et al. [116] found that the pretreatment with NaOH solution improved the surface roughness and hydrophobicity of CR, increased the adhesion between CR and cement mortar, and enhanced the blocking effect on sulphate attack. Liu et al. [40] pre-coated rubber particles with six modifiers, and all of them made the anticorrosion coefficients of RC larger than 97%. Synthetic resin is the best modifier, considering the effects of precoating on compressive strength and freeze–thaw resistance. However, RA replaces part of natural aggregate and blends into HSC reduces the resistance to sulphate attack, which may be attributed to the increased HSC water absorption with RA incorporation, making the aggressive solutions enter the internal structure easily [115].

## 7. Chloride Penetration Resistance

The rubber content and particle size effect the resistance of RC to chloride ion penetration. Table 10 demonstrates that rubber incorporated into concrete as a substitution for fine aggregate can effectively improve the chloride ion attack resistance. The rubber content increases causing chloride ion permeability decreases, owing to the impervious property of RA and the ability to increase the length and sinuosity of capillary channels [77]. However, excessive rubber content causes negative effects. Gheni et al. [118] replaced cement with rubber fibre powder (particle size < 0.075 mm) in five ratios of 5%, 10%, 15%, 20%, and 25%. When the content was less than 20%, the charge passed decreased, as the rubber powder content increased. When the amount exceeded 20%, the charge passed greatly increased. Thomas et al. [110] observed that the optimum amount of CR to replace fine aggregate is 7.5% for chloride penetration resistance. This condition can be attributed to the lack of compactness when excessive RA are mixed into concrete, leading to significant chloride penetration; it is similar to the reason for water absorption and permeability of RC [69,75]. Gupta et al. [107] described that rubber fibre (width: 2–5 mm, length: 20 mm) with hydrophobic nature and increasing crookedness and tortuosity acts as a good blocking mechanism between cement and chloride ion. Due to the small particle

size, rubber powder (0.15–1.9 mm) can fill the internal pores of concrete, which help to enhance the resistance to chloride ion attack [99]. However, Fernández-Ruiz et al. [119] found that the rubber and cement lacked good cementing ability. When small particles of rubber powder (0.063–0.6 mm) were used in place of cement in concrete, the gap between them became a seepage channel, and increased rubber powder content led the chloride permeability coefficient to increase. The same phenomenon was observed when rubber powder (0–0.3 mm) was incorporated into SCC [77].

As shown in Table 10, several experimental studies have attempted to treat rubber particle surface with chemicals, such as NaOH, SCA, or CSBR latex, to enhance the RA surface adhesion ability. Mineral admixtures, such as SF or FA, are incorporated into RC to reduce internal porosity, thereby enhancing the resistance to chloride ion penetration. SF can effectively fill the pores at the joint of aggregate and cement, making the interior of concrete highly compact [47,68,120,121]. Dong et al. [122] found that pretreated CR particles with SCA and precoated with cement paste, the chloride ion penetration resistance can be improved significantly. As shown in Figure 12, after pretreating with SCA, the coupling agent can attach well on the rubber surface, which can make the cement matrix adhere to rubber particle better. Guo et al. [39] used NaOH or SCA to pretreat the surface of rubber particles and precoated the rubber particles with normal cement, SF blended with cement, or cement plus $Na_2SiO_3$. All of these measures could greatly increase electrical resistivity, which meant low chloride permeability of RC. NaOH treatment could reduce ITZ porosity, and SF could refine internal pores. After treatment with SCA, the rubber surface became hydrophilic, and it is easier to form hydrogen bonds on a rubber surface, thereby benefiting the cementation between them. After treatment with CSBR, a chemical bond amongst rubber particles was formed. The enhanced interfacial adhesion prevented the rubber particles from floating due to vibration during the rubber-concrete pouring process. Compared with RC without pretreatment, rubber pretreated with SCA and CSBR could be more uniformly dispersed in concrete [33,123,124].

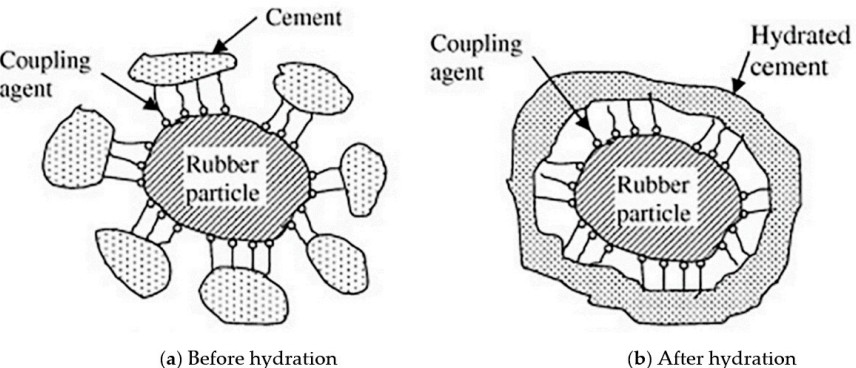

(**a**) Before hydration  (**b**) After hydration

**Figure 12.** The condition of RA surface after two step surface treatment [122].

**Table 10.** Influence of rubber on the concrete chloride penetration resistance.

| Reference | Treatment Method | RA Type and Size (mm) | RA Replacement Ratio (%) | Replacement Pattern | Concrete Type | Chloride Penetration Resistance | D [a], CD, CP, ER Compared to the Control Type |
|---|---|---|---|---|---|---|---|
| Gheni et al. [118] | Untreated | Rubber fibre powder: <0.075 | 5, 10, 15, 20, 25 by volume | Cement | OC | ↑ [b] | 20% ↓, 40% ↓, 50% ↓, 75% ↓, 200% ↑ (CD) |
| Thomas et al. [110] | Untreated | CR: 2–4 (25%) + 0.8–2 (35%) and rubber powder: 0.6 (40%) | 2.5, 5, 7.5, 10, 12.5, 15, 17.5, 20 by volume | FAG | OC | ↓ | 7.7% ↓, 7.7% ↓, 7.7% ↓, 0 ↑, 7.7% ↑, 23.1% ↑, 23.1% ↑, 30.8% ↑ (D) |
| Al-Akhras and Smadi [99] | Untreated | Rubber ash: 0.15 | 5, 10 by volume | FAG | OC | ↑ | 72.27% ↓, 81.33% ↓ (CP) |

**Table 10.** *Cont.*

| Reference | Treatment Method | RA Type and Size (mm) | RA Replacement Ratio (%) | Replacement Pattern | Concrete Type | Chloride Penetration Resistance | D [a], CD, CP, ER Compared to the Control Type |
|---|---|---|---|---|---|---|---|
| Fernández-Ruiz et al. [119] | Untreated | Rubber powder: 0.063–0.6 | 2.5, 5, 10 by volume | Cement | OC | ↓ | 5.62% ↑, 9.68% ↑, 21.58% ↑ (CD) |
| Bravo and Brito [69] | Untreated | CR: <11.2 | 5, 10, 15 by volume | FAG | OC | ↑ | 18.67% ↓, 7.33% ↓, 6.67% ↑ (CD) |
| Gupta et al. [107] | Untreated | Rubber powder: 0.15–1.9 | 5, 10, 15, 20 by volume | FAG | OC | ↑ | 7.32% ↓, 8.54% ↓, 14.63% ↓, 24.39% ↓ (CD) |
| Sagawa et al. [110] | Untreated | CR: 1–3 | 10, 15, 20 by volume | FAG | OC | ↑ | 1.61% ↓, 3.46% ↓, 22.35% ↓ (CD) |
| Li et al. [77] | Untreated | CR: 1–2, 0–0.3 | 30 by volume | FAG | OC | ↑ | 21.44% ↓, 12.10% ↓(CP) |
| Thomas et al. [125] | Untreated | CR: 2–4 (25%) + 0.8–2 (35%) and rubber powder: 0.6 (40%) | 2.5, 5, 7.5, 10, 12.5, 15, 17.5, 20 by volume | FAG | HSC | ↓ | 6.25% ↓, 6.25% ↓, 6.25% ↓, 0, 12.5% ↑, 18.75% ↑, 18.75% ↑, 25% ↑ (D) |
| Hall and Najim [85] | Untreated | CR: 2–6 | 44 by volume | FAG and CAG | SCC | ↓ | 152% ↑ (CD) |
| Oikonomou and Mavridou [123] | Untreated | CR: 0.09–1 | 2.5, 5, 7.5, 10, 12.5, 15 by weight | FAG | OC | ↑ | 14.22% ↓, 16.76% ↓, 25.43% ↓, 30.25% ↓, 35.18% ↓, 35.85% ↓ (CP) |
| Gesoğlu and Güneyisi [75] | Untreated | CR: 0.2–3 | 5, 15, 25 by volume | FAG | SCC | ↓ | 9.09% ↑, 13.64% ↑, 40.91% ↑ (CP) |
| Dong et al. [122] | Pre-treating with SCA and pre-coating with cement. | CR: 0.6–4.75 | 15, 30 by volume | FAG | OC | ↑ | 26.9% ↓,13% ↓ (CD) |
| | Pre-treating with NaOH. | CR: 1.5–2.8 | 15, 25, 35, 50 by volume | FAG | OC | ↑ | 35.71 ↑, 50.00 ↑, 55.71 ↑, 51.43 ↑ (ER) |
| | Pre-treating with SCA and pre-coating with cement. | CR: 1.5–2.8 | 15 by volume | FAG | OC | ↑ | 57.41 ↑ (ER) |
| Guo et al. [39] | Pre-treating with NaOH then pre-coating with cement. | CR: 1.5–2.8 | 15 by volume | FAG | OC | ↑ | 50.00 ↑ (ER) |
| | Pre-treating with NaOH then pre-coating with SF and cement. | CR: 1.5–2.8 | 15 by volume | FAG | OC | ↑ | 71.43 ↑ (ER) |
| | Pre-treating with NaOH then pre-coating with $Na_2SiO_3$ and cement. | CR: 1.5–2.8 | 15 by volume | FAG | OC | ↑ | 52.86 ↑ (ER) |
| Li et al. [33] | Pre-treating with SCA and CSBR. | Rubber powder: <0.6 | 5, 10, 15, 20, 30 by volume | FAG | OC | ↑ | 35.84% ↓, 34.09% ↓, 9.15% ↓, 2.83% ↑, 16.34% ↑ (CD) |

[a] Abbreviations: D: Depth of chloride penetration, CD: Coefficient of chloride diffusion, CP: Charge passed, ER: Electrical resistivity. [b] Evaluate standard change compared to the control type (%), corresponding to RA replacement ratio. Increase: ↑, Decrease: ↓.

## 8. Carbonation Resistance

Table 11 shows that RA increases the depth of carbonation by replacing natural aggregates incorporated into concrete. Gheni et al. [118] supposed that carbonation depth is closely related to concrete permeability, which is related to the distribution and size of internal pores; as the exposure time to the $CO_2$ environment increases, the carbonation depth increases [49]. Concrete workability deteriorates due to rubber incorporation; the interior of concrete is difficult to compact during mixing, consequently forming internal pores and cracks that make it easy for $CO_2$ to penetrate the concrete, and replacing coarse aggregate with rubber results in greater carbonation depth compared with replacing fine aggregate [69]. Thomas and Gupta [126] found that the reason for reduction in carbonation resistance is similar to that for reduction in permeability resistance. Both are attributed to the hydrophobic nature of RA, causing the cement matrix to be hard to adhere to the rubber particle surface stably. As a result, internal pores and cracks increase, which becomes a wide channel for $CO_2$ to penetrate into concrete. Pham et al. [113] found that after the CR was pretreated with NaOH, the adhesion was improved and the penetration of $CO_2$ was inhibited. In addition, the NaOH remaining on the rubber reacted with $CO_2$, which was conducive to reducing the carbonation depth.

**Table 11.** Influence of rubber on the concrete carbonation resistance.

| Reference | Treatment Method | RA Type and Size (mm) | RA Replacement Ratio (%) | Replacement Pattern | Carbonation Resistance | Carbonation Depth Compared to the Control Type |
|---|---|---|---|---|---|---|
| Gheni et al. [118] | Untreated | Rubber powder: <0.075 | 5, 10, 15, 20, 25 by volume | Cement | ↓ [a] | 50% ↑, 75% ↑, 150% ↑, 50% ↑, 250% ↑ |
| Thomas et al. [125] | Untreated | CR: 2–4 (25%) + 0.8–2 (35%) and rubber powder: 0.6 (40%) | 2.5, 5, 7.5, 10, 12.5, 15, 17.5, 20 by volume | FAG | ↓ | 0, 9.09% ↓, 9.09% ↓, 9.09% ↓, 0, 9.09% ↑, 18.18% ↑, 27.27% ↑ |
| Gupta et al. [49] | Untreated | Rubber fibres: 2–3 width and 20 in length | 5, 10, 15, 20, 25 by volume | FAG | ↓ | 2.68% ↑, 11.61% ↑, 15.18% ↑, 20.54% ↑, 25.00% ↑ |
| Bravo and Brito [69] | Untreated | CR: 1 | 5, 10, 15 by volume | FAG | ↓ | 14.29% ↑, 21.43% ↑, 42.86% ↑ |
| Pham et al. [113] | Pre-treating with NaOH | CR: 1–7 | 15, 30 by volume | FAG and CAG | ↑ | 28.2% ↓, 16.7% ↓ |

[a] Evaluate standard change compared to the control type (%), corresponding to RA replacement ratio. Increase: ↑, Decrease: ↓.

However, Thomas et al. [125] observed that with 2.5–12.5% rubber content, compared to the control group, the carbonation depth was smaller; when it was more than 12.5%, the depth of carbonation increases with increasing rubber content. When the content was 2.5–12.5%, the substituted fine aggregate had a similar particle size to rubber particles, causing the RC easier to mix, and the aggregate distribution inside was more uniform, which made the RC compact after mixing. Excessive rubber content leads to increased internal pores and lack of filling [113].

## 9. Alkali–Silica Reaction Damage Resistance

Previous research demonstrated that in concrete, using RA to replace partial nature aggregate can alleviate the damage caused by ASR. ASR occurs between the $OH^-$ and active silica aggregate in the alkaline environment. It generates a hygroscopic gel in the pore of cement matrix, which causes internal pressures and structure rupture with gel volume expansion [127,128]. RA has lower elastic modulus than natural aggregate, and has good deformation ability, which can absorb the stress created by ASR gel volume expansion. Therefore, ASR damage can be alleviated to some extent [110]. Afshinnia and

Poursaee [129] found that replacing natural fine aggregate with 16% and 24% RAs reduced the expansion of ASR by 43% and 39%, respectively. Rubber particles do not react in an alkaline environment and can prevent the development of cracks and dissipate the energy that caused cracks; as a result, fewer cracks and swelling were observed in RC compared with the control group [130]. The incorporation of synthetic polypropylene (PP) fibre into RC could help alleviate the swelling stress and reduce the ASR damage, The PP fibre have the bridging effect, which can maintain internal integrity [102]. After adding rubber (15%) and PP fibre (0.5%), the swelling caused by ASR was less than 0.1% at 14 days, which means very low possibility of ASR damage according to ASTM C1260 [131].

Pretreatment of the CR surface with NaOH solution, and then mixing the treated CR into self-compacting mortar (SCM) can alleviate the expansion of ASR [95]. ASR expansion of normal SCM and rubber-modified SCM is shown in Figure 13. The internal cracks of RC were reduced because the NaOH treatment strengthened the binding ability of cement matrix and CR. This condition resulted in a decrease in the mobility of alkali and water and helped reduce ASR gel. However, excessive rubber content reduced the stiffness of concrete and caused great deformation.

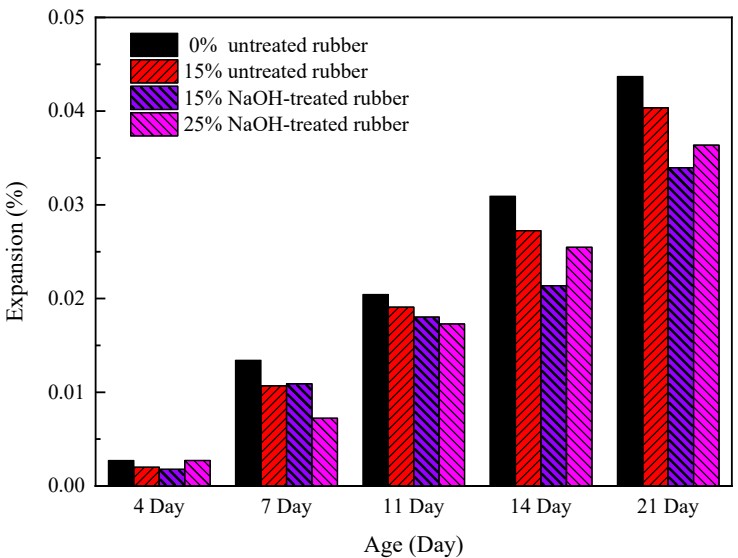

**Figure 13.** ASR expansion of different types of SCM at different ages [89].

## 10. Conclusions

This work introduces approximately 90 research results found in the past 10 years, involves various durability of RC and treatment methods of rubber, the following conclusions are drawn:

1. Pre-treating and pre-coating of rubber can reduce the internal porosity of RC, the occurrence and development of cracks at ITZ of RC, and thus enhance the durability of RC. When selecting treatment materials, comprehensive consideration should be given to the effect of improving durability, the feasibility of operating procedures, cost consumption, and environmental impact. The pretreating and precoating processes have a promoting significance for the application of rubber to concrete structures.

2. Rubber particles reduce the abrasion resistance of concrete. There are two main reasons for reducing RC abrasion resistance, one is high porosity and the other is weak adhesion on the RA surface, and a rubber content of 5–10% has a slightly negative effect on abrasion resistance. By contrast, adding SCMs or pre-treatment of rubber can effectively improve the abrasion resistance. In addition, well-graded rubber particles contribute to improved abrasion resistance.

3. The non-hydrophilic nature properties of RA leading the poor binding ability, which make it easy to form pores and water seepage channel in RC, these interconnected

channels help to increase the water absorption and permeability. Well-graded rubber particles can make RC denser than a single particle size. When the rubber content is less than 10%, the increase in water absorption is small or even decreases. When the rubber content is too large (greater than 15%), the water absorption increases significantly. Rubber with a small particle size (0–1 mm) can effectively fill the pores and water seepage channels, which can effectively reduce the water absorption and enhance the RC impermeability. Pre-treatment of rubber particles can effectively reduce water absorption and impermeability of RC.

4.  The pores created by the rubber incorporated into the concrete can play a role in stress absorption, thereby enhancing the freeze–thaw resistance of concrete. The best freeze–thaw resistance is achieved when the rubber content is 25–30%. RC with small-sized rubber particles (0–1 mm) has high freeze–thaw resistance. Pretreating rubber with NaOH and precoating it with synthetic resin and styrene–butadiene-type copolymer can significantly enhance the RC freeze–thaw resistance. The rubber replaces partial natural aggregate in concrete can enhance the freeze–thaw resistance but negatively affect the concrete strength. RC is suitable for areas without high-strength requirements but with high freeze–thaw resistance requirements.

5.  Using rubber particles in ordinary concrete can enhance the acid resistance, and the effect of well-graded rubber particles and fibre is significant. Rubber particles decrease concrete strength, 5–15% rubber content can meet the requirements of use strength and has high resistance to acid attack. Rubber can effectively improve the resistance to sulphate attack. Compared with concrete without RA, the RC has better deformation ability. Rubber can relieve internal expansion stress caused by sulphate attack. The best sulphate attack resistance is obtained when the rubber content is 5–10%. Synthetic resin is a good modifier for precoated rubber particles considering compressive strength change, resistance to sulphury acid, and freeze–thaw.

6.  The water absorption resistance and impermeability have a closely relation with the chloride penetration resistance. The addition of an appropriate amount of rubber (5–20%) to concrete can effectively improve the resistance to chloride permeability. The effect of RA replaces fine aggregate is better than that of coarse aggregate. Rubber fibre and fine rubber particles have a better effect on enhancing resistance to chloride attack. Rubber particle size should not be larger than 3 mm. Pre-treating rubber particles with NaOH or SCA and pre-coating rubber with CSBR latex, $Na_2SiO_3$-mixed cement paste, or SF-mixed cement can effectively enhance the resistance to chloride ion penetration.

7.  The addition of rubber particles to concrete increases carbonation depth. The principle of enhancing the carbonation resistance of RC is similar to that of reducing water absorption. Pre-treatment increases the adhesion of the rubber and increases the density of the RC, thereby inhibiting $CO_2$ penetration and effectively reducing the depth of carbonation.

8.  Rubber particles replace partial nature aggregate can alleviate the internal structure damage, caused by ASR. RA do not react in an alkaline environment and have good deformation ability; ASR gel expansion stress happening in concrete internal structure can be alleviated by flexible RA, which can prevent cracks from developing and dissipate the energy that caused cracks. Pretreating rubber with NaOH can enhance the resistance of RC to ASR damage. However, excessive rubber content (greater than 25%) leads to the deformation of RC, which is unbeneficial to structural services.

9.  Through the analysis of the rubber particle size, replacement ratio, and replacement pattern, the recommended as follows: the replacement pattern FAG is preferred, followed by cement material and CAG, and the rubber particle size and replacement ratio are 0–3 mm and 5–20%, respectively. If auxiliary cementitious materials are added, the replacement ratio of rubber can be appropriately increased by 5–10%.

## 11. Further Research Needs

1.  The treatment methods of rubber particles should be further studied, especially the related research of physical-chemical coupling treatment, so as to increase the bonding between rubber and cement-based, and then improve the durability.
2.  More research on the improvement of concrete durability by rubber fiber can be carried out.
3.  Combined with microstructure analysis, more research can be carried out on the durability of RC, especially the long-term durability.
4.  Further research is needed on RC ductility and energy absorption.
5.  The insulation, sound insulation, thermal resistance, and corrosion resistance of RC need to be further studied.

**Author Contributions:** Conceptualization, Y.L. and R.W.; methodology, Y.L.; software, J.C.; validation, Y.L., R.W. and Y.Z.; formal analysis, J.C.; investigation, X.T.; resources, Y.L.; data curation, J.C. and Y.Z.; writing—original draft preparation, J.C.; writing—review and editing, Y.L.; visualization, J.C.; supervision, Y.L.; project administration, R.W.; funding acquisition, Y.L. All authors have read and agreed to the published version of the manuscript.

**Funding:** This research was funded by Project funded by China Postdoctoral Science Foundation (2020M683687XB and 2022T150525), the National Natural Science Foundation of China (52009110 and 51879217), Natural Science Basic Research Program of Shaanxi (2021JM-331), Young Talent Fund of Association for Science and Technology in Shaanxi, China (20220416), and Young Talent fund of Association for Science and Technology in Xi'an City (095920211332).

**Institutional Review Board Statement:** Not applicable.

**Informed Consent Statement:** Not applicable.

**Data Availability Statement:** Not applicable.

**Conflicts of Interest:** The authors declare no conflict of interest.

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
