# Peer review of "A Review of the Durability-Related Features of Waste Tyre Rubber as a Partial Substitute for Natural Aggregate in Concrete"

_buildings, doi:10.3390/buildings12111975_

Round 1
Reviewer 1 Report
Manuscript Number: buildings-1965304-peer-review-v1
Reviewer’s Comments
The author(s) reviewed the factors that affect the durability of rubber concrete, such as the rubber replacement ratio, the replacement pattern, the particle size, and the rubber treatment method. The scholars reported that the increases in rubber content improve concrete's resistance to damage from chloride ions, acids, sulfates, temperature extremes, the alkali-silica reactions, and the optimal rubber content for that purpose is between 5-20%. Moreover, the ideal solution for durability is rubber replacing fine aggregate, followed by cement and coarse aggregate; additionally, the recommended rubber particle size is 0-3 mm. Furthermore, the author(s) reported that rubber particles have a negative impact on abrasion resistance, impermeability, water absorption resistance, and carbonation resistance. And the rubber particle surface pretreatment and incorporation of mineral admixtures into rubber concrete were effective methods to improve rubber concrete durability. The reviewer appreciates the idea presented in the manuscript. However, the reviewer has the attached comments that require Major revision.

Reviewer 2 Report
The authors included 122 papers to conduct a review on waste tyre rubber as replacement of aggegate in concrete. The authors focused on durability properties such as freezea nd thaw, water absorption, permeability, abrasion and etc. The paper deserves to be published. However, there are some major issues before publication:
A general discussion should be added to introduction. The importance of recycling, sustainable concrete should be indicated in the first paragraph of introduction. The following studies should be used for this purpose: Concrete Containing Waste Glass as an Environmentally Friendly Aggregate: A Review on Fresh and Mechanical Characteristics; Performance assessment of fiber-reinforced concrete produced with waste lathe fibers; Improvement in Bending Performance of Reinforced Concrete Beams Produced with Waste Lathe Scraps
Waste tyres can be utilized in two ways: rubber and steel wire. The authors should indicate this using following study: Investigation improvement in shear performance of reinforced concrete beams produced with recycled steel wires from waste tyre
How did papers conducted abrasion resistance test?
There are different ways of freezing thraw such as water frezing or air frezing, air thraw water. Please indicate these in tables. Also different tempratures were applied. This should be also indicated.
Suggestion of use of rubber in percentage depending on the tests can be provided
Round 2
Reviewer 1 Report
Dear author(s),
Thanks for doing the required corrections. I don't have further comments.
Reviewer 2 Report
The authors completed the request of the reviewer.